# Walking Promotion in Pregnancy and Its Effects on Insomnia: Results of Walking_Preg Project (WPP) Clinical Trial

**DOI:** 10.3390/ijerph191610012

**Published:** 2022-08-13

**Authors:** Rebeca Benito-Villena, Ingrid Guerrero-Martínez, Mariña Naveiro-Fuentes, Naomi Cano-Ibánez, Pedro Femia-Marzo, José Luis Gallo-Vallejo, Juan Mozas-Moreno, Carmen Amezcua-Prieto

**Affiliations:** 1Obstetrics and Gynecology Service, Hospital Universitario Virgen de las Nieves, 18014 Granada, Spain; 2PhD Program in Clinical Medicine and Public Health, University of Granada, 18016 Granada, Spain; 3Department of Preventive Medicine and Public Health, Faculty of Medicine, University of Granada, 18016 Granada, Spain; 4Consortium for Biomedical Research in Epidemiology and Public Health (CIBERESP), 28029 Madrid, Spain; 5Instituto de Investigación Biosanitaria (ibs.Granada), 18014 Granada, Spain; 6Unit of Biostatistics, Department of Statistics and O.R., Faculty of Medicine, University of Granada, 18016 Granada, Spain; 7Department of Obstetrics and Gynecology, Universidad de Granada, 18016 Granada, Spain

**Keywords:** pedometer, walking, pregnancy, Insomnia

## Abstract

**Introduction:** Insomnia is a frequent condition during pregnancy. The aim of this study was to assess if a walking promotion program from the 12th Gestational Week (GW) of pregnancy helps to prevent insomnia and improve the quality of sleep at third trimester. **Materials and Methods**: A prospective, randomized, and controlled trial was conducted with 270 pregnant women divided into 3 groups in parallel: maximum intervention group, I1 (pedometer and goal of 10,000 steps/day), minimum intervention group, I2 (pedometer without a goal), and control group (no intervention). All groups received recommendations about physical activity in pregnancy. A structured interview was performed at 13th, 20th, and 32nd GW, collecting pedometer mean steps/day, Athens Insomnia Scale (AIS), and Pittsburgh questionnaire (PSQI). Lineal regression models were conducted to determine the association between mean steps/day at 31st GW and AIS or PSQI score. **Results:** At 19th GW, groups I1 and I2 reached a mean of 6267 steps/day (SD = 3854) and 5835 steps/day (SD = 2741), respectively (*p* > 0.05). At 31st GW mean steps/day was lower for I2 (*p* < 0.001). Insomnia and poor sleep quality prevalence increased through pregnancy, but no differences between groups, within trimesters, were found (*p* > 0.05). Lineal regression showed no association between the average steps/day at third trimester of pregnancy and AIS and PSQI scores. **Conclusions:** Our walking promotion program based on pedometers did not help to prevent insomnia in the third trimester of pregnancy.

## 1. Introduction

Insomnia is defined as the reduction of sleep time and/or its quality that occurs despite appropriate circumstances, which leads to a clinically significant situation and/or interferences with daily activities [1]. Insomnia is related to cardiovascular disease, neuroendocrine disorders, pro-inflammatory states, and mental health disorders [2,3,4]. Sleep disturbances are more frequent in pregnant women compared with non-pregnant due to anatomical, physiological, hormonal, and psychological changes that occur in this stage of a woman’s life [2,3,4,5]. Insomnia is related to multiple negative pregnancy outcomes, such as gestational diabetes mellitus, gestational hypertensive disorders, preterm birth or prolonged second stage of labour [4,5,6]. 

Although drugs are the most common treatment for sleep disturbances, they should be avoided or only used for a short time during pregnancy because of the potential foetal harm. Hence, non-pharmacological treatments are very appealing for pregnant women [4,7]. 

Multiple studies have shown that physical activity has a positive impact on sleep continuity and quality [2]. Nevertheless, few studies have researched this association during pregnancy and their results are inconsistent [8]. Some randomised clinical trials have shown a positive association between duration and/or quality of sleep and water exercise [9], yoga [10], and aerobic exercise programmes [11]. 

The last World’s Health Organization (WHO) published guideline of physical activity includes a specific section for pregnancy where it recommends pregnant women to do at least 150 min of moderate-intensity physical activity throughout the week, emphasising the relevance of aerobic and muscle-strengthening activities [12].

Though walking is one of the most practised physical activities during pregnancy and one of the most recommended by health care professionals [13], to date the relationship between walking and insomnia in pregnancy has not been examined. The aim of this study was to assess if a walking promotion program in the first and second trimester of pregnancy could prevent insomnia and improve the quality of sleep in the third trimester of pregnancy. 

## 2. Materials and Methods

The complete study protocol of this RCT study was previously published [14]. This paper summarizes the essential information of the methodology and how sample size was determined—for further information, please check the study protocol.

A prospective, randomized controlled trial was conducted at the Obstetrics Unit of the Virgen de las Nieves Mother and Child Hospital, a third level public hospital that serves a reference population of 439,035 inhabitants in Granada (Andalusia, Spain). 

Ethical approval for the planned study was obtained from Research Ethics Committee in February 2019 (Peiba 1644-N-18). The trial was registered with the U.S. National Library of Medicine Trials registry (www.ClinicalTrials.gov identifier NCT03735381).

### 2.1. Selection Criteria for Elegibility

Inclusion criteria (age between 18 and 39 years, low risk pregnancies, sedentary women, available mobile and e-mail) and exclusion criteria (chronic disease, relative or absolute repose needed, active women, insomnia at the beginning of pregnancy or taking drugs due to sleep problems, intellectual deficits, or difficulty to understand the language) have been published expanded in the protocol [14].

### 2.2. Intervention

Women were randomized into three intervention groups. Group 1 (I1)—*maximum intervention*: pedometer + goal of steps/day to be reached (10,000 steps/day) + reminders (phone reminding the goal to achieve and a notification that the researching staff will proceed to collect the average count of steps/day of the week prior to 20th and 32nd GW); group 2 (I2)—*minimum intervention*: pedometer without goal nor reminders; and Control group (C)—*no intervention*. In addition, all the groups received information about recommendation of physical activity during pregnancy, according to The American College of Obstetricians and Gynaecologists (ACOG) [15].

The pedometer used was the Xiaomi Mi Band 2™ (Foxconn, Beijing, China), which was synced to patients’ personal smartphones. Patients had three meetings with a member of the research team who conducted a structured interview at 12th, 20th, and 32nd GW. 

### 2.3. Collected Variables

All variables were collected by a trained healthcare worker. Baseline variables were collected only in the first interview [14]. 

Outcome variables were: -Change in mean steps/day after intervention in the study arms, using pedometer register, at 19th and 31st GW;-Insomnia prevalence in every trimester of pregnancy, assessed by the Athens Insomnia Scale (AIS) [16], considering insomnia from a score equal to or greater than 7 points;-Quality of sleep in every trimester of pregnancy: assessed by the Pittsburgh scale (PSQI) [17]: a PSQI score of >5 indicates poor sleep quality.

### 2.4. Location and Data Collection

Participants were enrolled by clinicians at the hospital service. A health care assistant, and interviewer, assigned women to interventions. All women who met the inclusion criteria and signed the informed consent and loan certificate were enrolled in the study. The study sample was collected between June 2019 and December 2020 at their first obstetrical ultrasound (12th GW) at the hospital. Follow-up was completed in December 2021.

### 2.5. Randomization of Participants and Concealment

The Stata Ralloc program generated a randomization sequence by blocks of *n* = 9 and 3 arms per block (A, B and C). There was concealment for the people in charge of carrying out the evaluation and analysis of the data. Neither the participants nor the professionals who indicated the intervention were blinded. Participants knew they were selected for either a pedometer group or a non-pedometer group, but they did not know if the intervention was minimum or maximum. 

### 2.6. Data Analysis

Central tendency and dispersion were calculated for the numerical and absolute and relative frequencies for the qualitative variables. The normality of the variables was tested with the Kolmogorov–Smirnov test. 

For the contrast of the hypothesis of equality between group I1 and group I2, in terms of number of daily steps at 19th and 31st GW, student *t*-test between groups and *t*-test for relative samples within groups were applied.

To determine differences between quantitative variables (AIS and Pittsburgh scores) throughout the study period (first, second, and third trimester), one-way ANOVA comparison between groups and repeated measure ANOVA within group were used. Subsequently, to assess the comparison between the arm groups with respect to the frequency of insomnia and quality of sleep throughout the study period, Pearson’s Chi-square was applied. Cochran’s Q test was used for comparison within groups. 

Finally, lineal regression models were carried out to determine the association between mean steps/day at 31st GW and AIS score and mean steps/day at 31st GW and PSQI score. Models were adjusted by intervention group, AIS score in second trimester (in first model), and PSQI in second trimester (in second model). 

Data analysis was carried out by protocol with Stata/SE v15.0 statistical package and IBM SPSS Statistics for Windows v28.0 (IBM Corp., Armonk, NY, USA). All contrasts were bilateral and statistical significance was set when *p* value was ≤0.05 for all analyses.

## 3. Results

A total of 285 pregnant women were assessed for eligibility. Of these, 15 were excluded because they refused or did not have time to walk. Therefore, there were 270 pregnant women randomized into 3 study arms: maximum intervention group I1 (*n* = 120), minimum intervention group I2 (*n* = 120), and control group C (*n* = 30).

Finally, 236 women completed the follow up (*n* = 108 in group I1, *n* = 101 in group I2, and *n* = 27 in group C) and were analyzed (Figure 1).

### 3.1. Characteristics of the Sample 

The participants baseline characteristics are described in Table 1. No clinically significant baseline imbalance between groups was observed. The mean age was 31.92 years (SD = 4.92) in group I1; 31.91 (SD = 5.56) in group I2 and 31.7 (SD = 5.36) in group C. Gestational age at recruitment for the three comparison groups was 12 weeks. 50% of women from groups I1 and I2, and 66.7% in the control group were nulliparous. A total of 19.32% in group I1, and 24.17% in group I2 were obese or extremely obese. About 60% of the women from the three arms of the study had no prior history of abortions and over 90% of them were from Spain. 

### 3.2. Adherence to the WPP Intervention

The mean steps/day reached at 19th and 31st GW for groups I1 and I2 is shown in Table 2 and Figure 2. When comparing groups, there were no differences for mean steps/day at 19th GW (*p* = 0.347). At 31st GW, group I1 walked more than group I2 (*p* = 0.022). As presented in Figure 2, both groups decreased their mean steps/day along pregnancy. This decrease was only significant in group I2 (*p* < 0.001). 

### 3.3. AIS (Insomnia) and Pitssburhg (Sleep Quality) Scores through Pregnancy in WPP Arms

Table 3 shows insomnia (AIS) and sleep quality (Pittsburgh) mean scores in each trimester of pregnancy, in the three arms of the study. When comparing the three groups for every trimester, no differences were found in AIS and Pittsburg scores (*p* > 0.05). For I1 and I2, mean AIS and Pittsburg score significantly increased as pregnancy progressed (*p* < 0.001). For group C, the AIS score did not change throughout pregnancy (*p* = 0.211), but the Pittsburg score increased (*p* = 0.001).

### 3.4. Insomnia and Poor Quality Sleep Prevalence through Pregnancy in WPP Arms

As pregnancy progressed, both insomnia and poor quality of sleep significantly increased for groups I1 and I2 (*p* < 0.001). For group C, the insomnia prevalence did not change (*p* > 0.05), while poor sleep quality did (*p* < 0.001). Comparing the three groups, no differences were found in insomnia prevalence or poor quality of sleep in each trimester of pregnancy (*p* > 0.05) (Table 4). 

### 3.5. Walking Intervention Effect on Insomnia and Quality of Sleep in Third Trimester of Pregnancy

A lineal regression was performed to estimate the association between walking in the third trimester of pregnancy and insomnia/sleep quality in the third trimester of pregnancy (Table 5). The crude and adjusted models showed no association between the number of mean steps achieved in the third trimester of pregnancy and the insomnia scale score, AIS3 (*ß* = −7.02 × 10^−5^, 95% CI −3.01 × 10^−4^–0.00), nor between the number of average steps/days reached in the third trimester of pregnancy and sleep quality scale score, PSQI3 (*ß* = −9.66 × 10^−5^, 95% CI −2.16 × 10^−4^–0.00). In the adjusted models we observed a positive association between AIS2 and AIS3 (*ß* = 0.53, 95% CI 0.38–0.68) and between PSQI2 and PSQI3 (*ß* = 0.53, 95% CI 0.38–0.67).

## 4. Discussion

### 4.1. Main Results

Groups I1 and I2, who received a pedometer at 12th GW, had similar mean steps/day counts at 19th GW. At 31st GW, mean steps/day was significantly lower for group I2 than for group I1. The insomnia and poor quality of sleep prevalence arose all through pregnancy, but differences between groups for each trimester were not found. The lineal regression showed no association between the average steps/day reached at the third trimester of pregnancy and AIS and Pittsburg scores in this trimester.

### 4.2. Comparison with Other Studies

Poor sleep quality and insomnia in pregnancy are more prevalent as pregnancy progresses, possibly related to pregnancy physical symptoms or discomfort [18], or high pre-pregnancy BMI condition [19]. 

The relationship between sleep and physical activity (PA) has been assessed in a cross-sectional study on 1259 pregnant women [20]. Physical activity was not strongly associated with sleep quality or duration. However, higher occupational PA was associated with sleep duration [20]. This observation matches a secondary analysis from a RCT study in which 29 sedentary women in the second trimester were enrolled in an 8-week PA intervention pilot study and were randomly assigned to intervention or control group. PA levels were positively predictive (*p* = 0.037) of sleep onset latency and negatively predictive (*p* = 0.01) of sleep quality, demonstrating a negative effect of PA on sleep during pregnancy when measured daily [21].

In a more recent study carried out with 2443 participants recruited from the Shanghai Maternal-Child Pairs Cohort study, moderate PA level significantly reduced PSQI scores at both second and third trimester. At 32nd–36th GW, a high PA level significantly decreased the PSQI score, with a greater decline than a moderate PA level. Compared with pregnant women who maintained or reduced their PA levels in the third trimester, those who increased their PA levels significantly improved their sleep quality [8]. Accordingly, a study designed to evaluate the effect in pregnant women of a non-supervised aerobic exercise intervention on sleep quality have shown a significant improvement in sleep quality in the intervention group at 4 weeks post-test, which persisted through 12 weeks post-test. The intervention group received a 20-min, low impact, aerobic exercise video on DVD and were instructed to exercise at home at least three times per week in a period of three months. The control group received routine prenatal care [22]. 

There are no previous studies about the effect of a walking intervention program and the prevention of insomnia in the third trimester of pregnancy. However, walking intervention combined with practicing deep breathing exercise has been applied in women with insomnia problems [23]. In this study, 60 pregnant women complaining from insomnia, at a gestational age from 28 to 32 weeks were enrolled. The researchers instructed the pregnant women to walk for about half an hour per day for two weeks, together with practicing deep breathing exercise. Women were contacted weekly by telephone to confirm the performance of the exercises. There were statistically significant differences between all component of Insomnia Severity Index pre and post intervention (*p* < 0.001), except in the component of pregnant women worried about their current sleep problem.

### 4.3. Limitations and Strenghs

Our women cohort showed non-adherence to the walking intervention program, in both intervention groups I1 and I2. After seven intervention weeks (12th to 19th GW), the mean steps/day count was under 6300 in both groups. Furthermore, at 32nd GW the number of steps/days was even lower. This might explain the lack of association found between walking and insomnia. Remarkably, the mean daily steps count decreased more in group I2 than in group I1. The goal set and the reminders might have a role in this difference as seen in previous studies [24].

Moreover, the COVID-19 pandemic must be considered. A total of 19% of the women were affected by the COVID-19 lockdown. The lockdown lasted four months in Spain and all non-essential workers were forbidden to leave their residences. In addition, pregnant women within the essential workers were encouraged to take their maternity leave since the beginning of the pregnancy because they were considered a vulnerable group [25]. Besides the restriction of movement, the COVID-19 pandemic represented a major stressing factor, since a highly contagious and invisible virus suddenly transformed neutral situations (social interaction or touching one’s face) to perceived potentially ones [26]. Moreover, according to the Spanish National Institute of Statistics, 475,000 people have lost their job since the beginning of the pandemic [27]. An international study based on an anonymous online survey between April and June 2020, showed that 67% of the participants felt from moderate to high stress level [26]. Another international study found that rates of insomnia, anxiety, and depression were very high during the first wave of the COVID-19 pandemic [28]. Several studies that have specifically addressed the connection between stressful life events and clinical insomnia support the idea that stressful life events may be a trigger to insomnia [29].

Another limitation of the study was the loss of follow up of 12.6% of the patients. Even if the loss was similar in the three groups, this could have affected the magnitude of the association trending towards null.

It is worth highlighting the contribution of electronic devices such as the pedometer, to be able to objectively measure the number of steps [30,31]. However, more effective, and less compliant methods are needed to measure the number of steps required until further research of this type is carried out.

Regarding the strengths of the study, although we have not found association, this is a pioneering study in the literature in the approach to the effect of walking on insomnia and quality of sleep towards the end of it. 

### 4.4. Implication for Practice

We observed that insomnia and poor sleep quality at the second trimester of pregnancy were a risk factor of having insomnia and poor sleep quality at third trimester. A screening of insomnia or quality of sleep at the first half part of pregnancy and health promotion measures could prevent the higher incidence of insomnia at the end of pregnancy. 

## 5. Conclusions

Our walking promotion program based on pedometers did not prevent the insomnia in the third trimester of pregnancy. However, we must consider these results cautiously because adherence to the walking program was rather low, and many participants were affected by the COVID-19 pandemic. This first approach to the role of walking in the prevention of third trimester insomnia did not find association, although it may have pointed the path for further research. 

## Figures and Tables

**Figure 1 ijerph-19-10012-f001:**
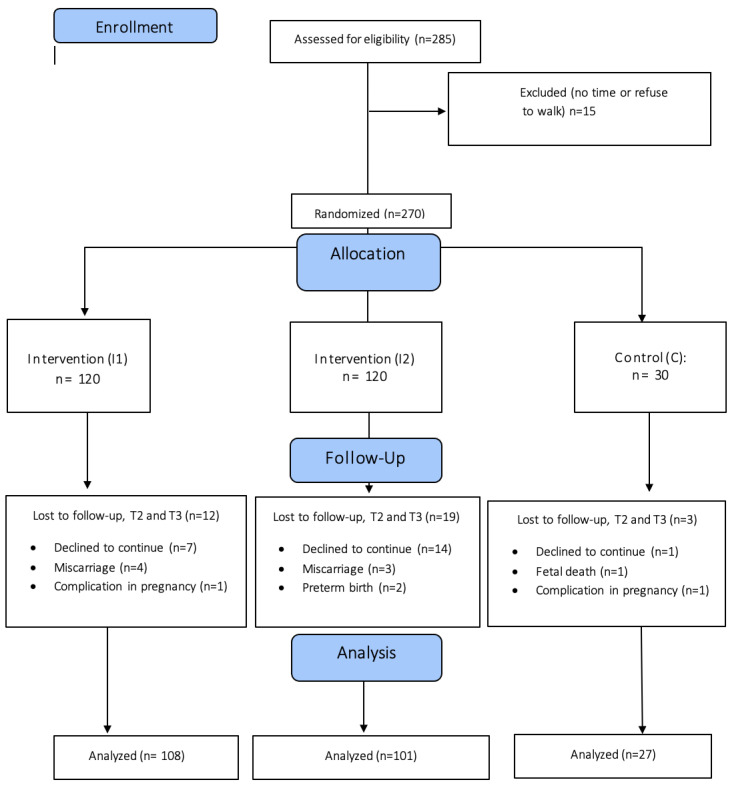
Flow chart. I1: maximum intervention group: pedometer, goal of 10,000 steps and reminder; I2: minimum intervention group: pedometer, without steps goal; C: control group: without pedometer, but with physical activity recommendations; T: Interview time; T2: second trimester; T3: third trimester.

**Figure 2 ijerph-19-10012-f002:**
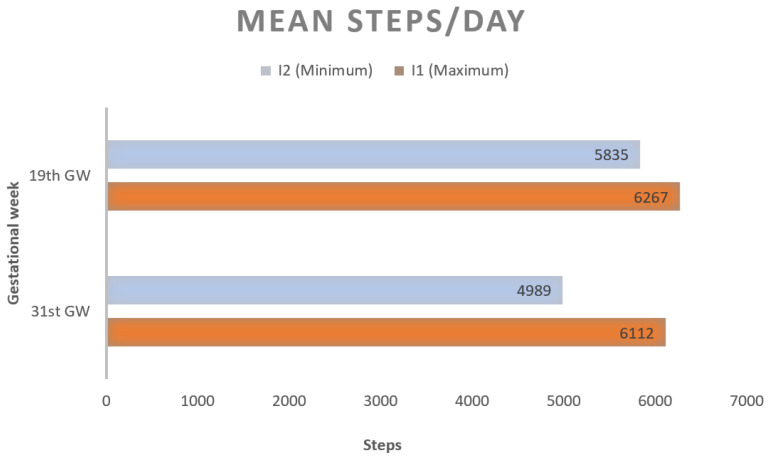
Mean steps/day at 19th and 31st GW.

**Table 1 ijerph-19-10012-t001:** Baseline characteristics of participants in the WPP.

Variable		Group I1 *n* = 120	Group I2 *n* = 120	Group C *n* = 30
Age; Mean (SD)		31.92 (4.92)	31.91 (5.56)	31.7 (5.36)
Gestational Week at recruitment; Mean (SD)		12.29 (0.063)	12.21 (0.71)	12.43 (0.73)
Previous children; N (%)	0	59 (49.58)	60 (50.00)	20 (66.67)
	1	43 (36.13)	40 (33.33)	8 (26.67)
	2	14 (11.76)	17 (14.17)	2 (6.67)
	3	3 (2.50)	3 (2.50)	0 (0.00)
Children under 3 years old; N (%)	Yes	8 (6.67)	6 (5.00)	1 (3.30)
	No	112 (93.33)	114 (95)	29 (96.67)
Previous abortions; N (%)	0	75 (62.50)	74 (61.67)	19 (63.33)
	1	33 (27.50)	32 (26.67)	9 (30.00)
	≥2	12 (10.00)	14 (11.67)	2 (6.67)
Nationality; N (%)	Spanish	108 (90.76)	109 (90.83)	29 (96.67)
	Other	11 (9.24)	11 (9.17)	1 (3.33)
Stable couple; N (%)	No	1 (0.84)	6 (5.00)	2 (6.67)
	Yes (non-cohabit)	1 (0.84)	2 (1.67)	2 (6.67)
	Yes (cohabit)	117 (98.32)	112 (93.33)	26 (86.67)
Social class; N (%)	I	14 (11.76)	23 (19.17)	3 (10.00)
	II	18 (15.13)	16 (13.33)	6 (20.00)
	III	31 (26.05)	20 (16.67)	5 (16.67)
	IV	39 (32.77)	46 (38.33)	12 (40)
	V	17 (14.29)	15 (12.50)	4 (13.33)
Pre-pregnancy BMI	Mean, SD	25.86 (5.30)	26.26 (5.76)	25.46 (6.28)
	Min–Max	18.05–46.71	17.82–48.74	17.15–40.58
Pre-pregnancy BMI; N (%)	Underweight (<18.5)	2 (1.68)	2 (1.67)	4 (13.33)
	Normal (18.5–24.9)	64 (53.78)	59 (49.17)	12 (40.00)
	Overweight (25–29.9)	30 (25.21)	30 (25.00)	8 (26.67)
	Obese (30–34.9)	14 (11.76)	21 (17.50)	3 (10.00)
	Extremely obese (>35)	9 (7.56)	8 (6.67)	3 (10.00)

Group I1 = maximum intervention group; group I2 = minimum intervention group; SD = Standard deviation; BMI= Body mass index; group C = control group.

**Table 2 ijerph-19-10012-t002:** Change in mean steps/day after intervention.

	Group I1 (Max)	Group I2 (Min)	Steps Difference	p^1^
Steps in 19th GW				
Mean	6267	5835	432	0.347
(SD)	(3854)	(2741)		
Steps in 32nd GW				
Mean	6112	4989	1123	0.022
(SD)	(3954)	(2680)		
Steps difference	−155	−846		
p^2^	0.749	0.001		

I = Intervention from 12 to 32nd GW; group I1 = maximum intervention group; group I2 = minimum intervention group; GW = gestational week; SD = standard deviation; p^1^: *t*-test comparison between groups; p^2^: *t*-test comparison within groups in 19th and 32nd GW.

**Table 3 ijerph-19-10012-t003:** Mean score of insomnia and poor quality of sleep throughout pregnancy.

	AIS Score Mean (SD); Range			p^1^	PSQI Score Mean (SD); Range			p^1^
	Group I1	Group I2	Group C		Group I1	Group I2	Group C	
1st trimester of pregnancy	3.06 (3.90) range 0–21	3.37 (4.53) range 0–22	3.40 (4.67) range 0–16	0.206	7.89 (2.98) range 3–17	7.80 (2.77) range 3–15	8.50 (3.26) range 3–15	0.472
2nd trimester of pregnancy	3.22 (4.40) range 0–24	3.55 (5.14) range 0–23	4.35 (5.75) range 0–17	0.107	7.99 (2.53) range 3–15	8.36 (2.64) range 4–16	8.32 (2.65) range 4–14	0.893
3rd trimester of pregnancy	5.59 (5.42) range 0–22	6.86 (6.06) range 0–24	5.93 (5.71) range 0–18	0.519	9.55 (3.02) range 4–19	9.90 (2.83) range 4–19.5	10.36 (2.85) range 5–16	0.795
p^2^	<0.001	<0.001	0.211		<0.001	<0.001	0.001	

Group I1 = maximum intervention group; group I2 = minimum intervention group; group C = control group; AIS = Athens Insomnia Scale; PSQI = Pittsburgh sleep quality index; SD = standard deviation; p^1^: One-way ANOVA comparison between groups; p^2^: repeated measure ANOVA within group; AIS score 0–24; PSQI score = 0–21.

**Table 4 ijerph-19-10012-t004:** Prevalence of insomnia and poor quality of sleep throughout pregnancy.

	Insomnia N (%) (AIS ≥ 8)			p^1^	Poor Sleep Quality N (%) (PSQI ≥ 5)			p^1^
	Group I1	Group I2	Group C		Group I1	Group I2	Group C	
1st trimester of pregnancy	16 (13.6)	17 (14.3)	6 (20.0)	0.776	92 (76.7)	90 (75.0)	22 (73.3)	0.837
2nd trimester of pregnancy	19 (16.7)	20 (18.5)	7 (25.0)	0.434	101 (84.2)	104 (86.7)	25 (83.3)	0.823
3rd trimester of pregnancy	41 (37.6)	43 (42.2)	9 (32.1)	0.283	113 (94.2)	117 (97.5)	29 (96.7)	0.416
1st vs. 2nd; p^2^	0.446	0.574	0.695		0.165	0.010	0.165	
1st vs. 3rd; p^2^	<0.001	<0.001	0.239		<0.001	<0.001	0.001	
2nd vs. 3rd; p^2^	<0.001	<0.001	0.433		0.011	0.006	0.064	

Group I1 = maximum intervention group; group I2 = minimum intervention group; group C = control group; p^1^: Chi square comparison between groups; p^2^: Cochran’s Q test. Significance values have been adjusted using the Bonferroni correction for various tests; AIS: Athens Insomnia Scale; PSQI: Pittsburgh sleep quality index.

**Table 5 ijerph-19-10012-t005:** Lineal regression model between mean/steps day at 31st GW and insomnia/sleep quality at the third trimester of pregnancy.

		Insomnia in Third Trimester (T3)				Sleep Quality in Third Trimester (T3)		
		ß Coef.	95% CI	p		ß Coef.	IC 95%	p
Crude models	Steps T3	−7.02 × 10^−5^	(−3.01 × 10^−4^–0.0)	0.549	Steps T3	−9.66 × 10^−5^	(−2.16 × 10^−4^–0.00)	0.113
Adjusted models	Steps T3 *	0.7 × 10^−4^	(−1.4 × 10^−4^–2.8 × 10^−4^)	0.515	Steps T3 **	−8.93 × 10^−5^	(−1.98 × 10^−4^–0.00)	0.109
	AIS2	0.53	(0.38–0.68)	<0.001	PSQI2	0.53	(0.38–0.67)	<0.001
	Group 1_2	1.14	(−0.31–2.59)	0.123	Group 1_2	−8.73 × 10^−5^	(−8.4 × 10^−1^–0.67)	0.820

Lineal Regression Model. ß Coef.: regression coefficient; T3: Third trimester; AIS3: Athens Insomnia Scale in T3; * T3 insomnia model adjusted by AIS2 (AIS in T2) and Group 1_2 (I1 vs. I2); ** T3 sleep quality model adjusted by PSQI2 (PSQI in T2) and Group1_2 (I1 vs. I2).

## Data Availability

The data presented in this study are available on request from the corresponding author.

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
