# Peer review of "Walking Promotion in Pregnancy and Its Effects on Insomnia: Results of Walking_Preg Project (WPP) Clinical Trial"

_ijerph, 2022, doi:10.3390/ijerph191610012_

Round 1
Reviewer 1 Report
Specific comments:
23: …270 pregnant women were randomized…. I suggest moving above to the part of Materials and Methods
108: The pedometer used was the Xiaomi Mi Band 2™, which was synced to patients' personal smartphones….. It is not a validated method or generally used in measurements of this type of research because it is highly susceptible to disturbances. More or less controlled movements of the hand or wrist result in counting "false" steps that did not occur, so sitting and properly rotating the wrist, we can carry out several thousand steps in an hour. The method is very unreliable and only allows a rough estimate of the number of steps taken.
201-214: In my opinion whole part should be moved to the very beginning of part 2 (Materials…) as point 2.1 Characteristics of the sample…..
287: The discussion should be included following the literature on the subject, which in that case has the most substantial impact on insomnia, poor sleep quality, and sleep disorders in general in pregnant women. And then what activities and procedures should be implemented to reduce or eliminate this disorder. Otherwise, the reader of this manuscript remains at least disappointed with the substantive level of the work, which adds nothing else to the fact that walking at different distances during the week does not improve the sleep disorders mentioned above disorders.
331-332: Regarding the study's strengths, it is worth highlighting the contribution of electronic devices such as the pedometer to measure the number of steps objectively… As commented in 108 verses, using not validated equipment is a weakness rather than a strength. It can be suggested at this point that more effective and less compliant methods are needed to measure the number of steps required until further research of this type is carried out.
333-336: We have encouraged pregnant women the habit of walking throughout pregnancy…... the results of this study do not indicate this, and the recommendations to remain active during pregnancy are apparent, as is the pro-health effect associated with it, but we do not find a statistically significant connection that the more steps taken per week, the less sleep disturbance I the third trimester of pregnancy.
Reviewer 2 Report
Dear Authors,
Congratulations on the study and on the idea! It is a huge pity that the study groups did not adhere to the protocol, but this is the aspect of clinical studies that always need to be takn into account. However, that of course does not influence on the quality of the study. I have only some minor comments:
l .153-4: "Participants knew they were selected for either a 153 pedometer group or a non-pedometer group, but they did not know if the intervention 154 was minimum or maximum".- In general it is true, but technically, it is not the way of blinding the gorup notto know if the intervention is minimal or maximal. However, it is nof a flawn.
Table 1.- show the statistical results of the comparison between the groups- if they differ or not (I assume they do not, but it has to be showed)
l. 298-300 "Some studies have observed the relationship between physical activity and quality of life, 298 depression, and insomnia in pregnant women[18]. Other studies have assessed the rela- 299 tionship between sleep and physical activity[19]"- show the studies and add some more details; they did not assess walking influence, but you have to show a relationship between the theoretical bases and the idea of the study
General suggestions: I would advise you to start a cooperation with another study site and repeat the study, maybe increasing slighlty the minimal requirements for the "walking group" to facilitate seeing the differences. The idea is great and should be verified in broader studies in more propitious circumstances
Best regards
